# Combinatorial Bayesian Optimization
# with Random Mapping Functions to Convex Polytopes

**Jungtaek Kim**[1]        **Seungjin Choi**[2]        **Minsu Cho**[1]

[1]POSTECH, South Korea
[2]Intellicode, South Korea

## Abstract

Bayesian optimization is a popular method for solving the problem of global optimization of an expensive-to-evaluate black-box function. It relies on a probabilistic surrogate model of the objective function, upon which an acquisition function is built to determine where next to evaluate the objective function. In general, Bayesian optimization with Gaussian process regression operates on a continuous space. When input variables are categorical or discrete, an extra care is needed. A common approach is to use one-hot encoded or Boolean representation for categorical variables which might yield a combinatorial explosion problem. In this paper we present a method for Bayesian optimization in a combinatorial space, which can operate well in a large combinatorial space. The main idea is to use a random mapping which embeds the combinatorial space into a convex polytope in a continuous space, on which all essential process is performed to determine a solution to the black-box optimization in the combinatorial space. We describe our combinatorial Bayesian optimization algorithm and present its regret analysis. Numerical experiments demonstrate that our method shows satisfactory performance compared to existing methods.

## 1 INTRODUCTION

Bayesian optimization [Brochu et al., 2010, Shahriari et al., 2016, Frazier, 2018] is a principled method for solving the problem of optimizing a black-box function which is expensive to evaluate. It has been quite successful in various applications of machine learning [Snoek et al., 2012], computer vision [Zhang et al., 2015], materials science [Frazier and Wang, 2016], biology [González et al., 2014], and synthetic chemistry [Shields et al., 2021]. In general, Bayesian opti-mization assumes inputs (decision variables) are real-valued vectors in a Euclidean space, since Gaussian process regression is typically used as a probabilistic surrogate model and the optimization of an acquisition function is a continuous optimization problem. However, depending on problems, input variables can be of different structures. For instance, recent work includes Bayesian optimization over sets [Garnett et al., 2010, Buathong et al., 2020, Kim et al., 2021], over combinatorial inputs [Hutter et al., 2011, Baptista and Poloczek, 2018, Oh et al., 2019], or over graph-structured inputs [Cui and Yang, 2018, Oh et al., 2019].

It is a challenging problem to find an optimal configuration among a finite number of choices, especially when the number of possible combinations is enormous [Garey and Johnson, 1979]. This is even more challenging when the objective function is an expensive-to-evaluate black-box function. This is the case where inputs are categorical or discrete, referred to as *combinatorial Bayesian optimization*, which has been studied by Baptista and Poloczek [2018], Buathong et al. [2020] recently. However, the methods by Hutter et al. [2011], Baptista and Poloczek [2018] employ one-hot encoded or Boolean representations to handle categorical or discrete input variables, which often suffer from the curse of dimensionality.

In this paper we present a scalable method for combinatorial Bayesian optimization, alleviating combinatorial explosion. Motivated by earlier work on online decision making in combinatorial spaces [Rajkumar and Agarwal, 2014], we solve combinatorial optimization in a convex polytope, employing an injective mapping from a categorical space to a real-valued vector space. However, in contrast to the work [Rajkumar and Agarwal, 2014], we use a random projection to embed the categorical space into the real-valued vector space on which all essential process is performed to solve the combinatorial Bayesian optimization. We show ours achieves a sublinear regret bound with high probability.

**Contributions.**    We provide a more general perspective than the existing methods, by defining on a convex polytope,

*Accepted for the 38th Conference on Uncertainty in Artificial Intelligence* (UAI 2022).

a generalization of 0-1 polytope. Based on this perspective, we propose a combinatorial Bayesian optimization strategy with a random mapping function to a convex polytope and their lookup table, which is referred to as CBO-Lookup. Finally, we guarantee that our method has a sublinear cumulative regret bound, and demonstrate that our method shows satisfactory performance compared to other methods in various experiments.

## 2 RELATED WORK

Bayesian optimization on structured domains, which is distinct from a vector-based Bayesian optimization [Močkus et al., 1978, Jones et al., 1998], attracts considerable attention from Bayesian optimization community, due to its potential on novel practical applications such as sensor set selection [Garnett et al., 2010], hyperparameter optimization [Hutter et al., 2011, Wang et al., 2016], aero-structural problem [Baptista and Poloczek, 2018], clustering initialization [Kim et al., 2021], and neural architecture search [Oh et al., 2019]. This line of research discovers a new formulation of Bayesian optimization tasks, by defining surrogates and acquisition functions on structured domains.

Especially, Hutter et al. [2011], Wang et al. [2016], Baptista and Poloczek [2018] focus on a combinatorial space at which all available items are defined as discrete or categorical variables, and solve this combinatorial problem by formulating on a binary domain, because any combination on a combinatorial space is readily expressed as binary variables. Hutter et al. [2011] propose a method based on random forests [Breiman, 2001] to find an optimal algorithm configuration, where a combination is composed of continuous and categorical variables. Wang et al. [2016] suggest a high-dimensional Bayesian optimization method that is able to handle combinations with scaling and rounding. Baptista and Poloczek [2018] solve an optimization problem over a discrete structured domain with sparse Bayesian linear regression and their own acquisition function. These approaches (or the variants of these methods) are compared to our method in the experiment section. Dadkhahi et al. [2020] solve a combinatorial optimization problem with multilinear polynomials and exponential weight updates. In particular, the weights are updated using monomial experts' advice. Deshwal et al. [2021] propose an efficient approach to solving combinatorial optimization with Mercer features for a diffusion kernel.

To compute a kernel value over combinations, a kernel with one-hot encoding [Duvenaud, 2014] can be directly employed. Hutter et al. [2011], Feurer et al. [2015] show this approach performs well in the applications of hyperparameter optimization. However, as in the experimental results reported by Baptista and Poloczek [2018], it underperforms in the optimization tasks of combinatorial structures. Furthermore, in this work, we will employ a baseline with

Aitchison and Aitken (AA) kernel [Aitchison and Aitken, 1976], defined on a discrete space. This AA kernel is symmetric and positive definite [Mussa, 2013], which satisfies the requirements of positive-definite kernels [Schölkopf and Smola, 2002].

## 3 COMBINATORIAL BAYESIAN OPTIMIZATION

We consider the problem of minimizing a black-box function $f(\mathbf{c}) : \mathcal{C} \to \mathbb{R}$, where the input $\mathbf{c} = [c_1, \ldots, c_k]^\top$ is a collection of $k$ categorical (or discrete)[1] variables with each variable $c_i$ taking one of $N_i$ distinct values:

$$\mathbf{c}^\star = \arg\min_{\mathbf{c} \in \mathcal{C}} f(\mathbf{c}), \tag{1}$$

where the cardinality of the combinatorial space $\mathcal{C}$ is

$$|\mathcal{C}| = N_1 \times \cdots \times N_k = N. \tag{2}$$

Bayesian optimization is an efficient method for solving the black-box optimization (1), where a global solution $\mathbf{c}^\star$ is found by repeating the construction of a surrogate and the optimization of an acquisition function.

Bayesian optimization estimates a surrogate function $\widehat{f}$ in order to determine a candidate of global solution $\mathbf{c}^\dagger$,

$$\mathbf{c}^\dagger = \arg\max_{\mathbf{c} \in \mathcal{C}} a(\mathbf{c}; \widehat{f}(\mathbf{c}; \mathbf{C}, \mathbf{y})), \tag{3}$$

where $a$ is an acquisition function, $\mathbf{C}$ is previously observed input matrix $[\mathbf{c}_1, \ldots, \mathbf{c}_t] \in \mathcal{C}^t$, and $\mathbf{y}$ is the corresponding noisy output vector $[y_1, \ldots, y_t]^\top \in \mathbb{R}^t$;

$$y = f(\mathbf{c}) + \epsilon, \tag{4}$$

where $\epsilon$ is an observation noise. This formulation has two technical challenges in (i) modeling a surrogate function over $\mathbf{c}$ and (ii) optimizing an acquisition function over $\mathbf{c}$. We thus suggest a combinatorial Bayesian optimization method to solving this problem.

Instantaneous regret $r_t$ at iteration $t$ measures a discrepancy between function values over $\mathbf{c}_t^\dagger$ and $\mathbf{c}^\star$:

$$r_t = f(\mathbf{c}_t^\dagger) - f(\mathbf{c}^\star), \tag{5}$$

where $\mathbf{c}_t^\dagger$ is the candidate determined by Bayesian optimization at iteration $t$. Using (5), we define a cumulative regret:

$$R_T = \sum_{t=1}^{T} r_t, \tag{6}$$

and the convergence of Bayesian optimization is validated to prove the cumulative regret is sublinear, $\lim_{T \to \infty} R_T / T = 0$ [Srinivas et al., 2010, Chowdhury and Gopalan, 2017].

---

[1] Only nominal variables are considered in this paper.

A widely-used method to handle these categorical variables in Bayesian optimization is to use one-hot encoded representation, converting $\mathbf{c}$ into a vector of length $N$ with a single element being one and the remaining entries being zeros [Duvenaud, 2014, Hutter et al., 2011, Wang et al., 2016]. In such a case, we need a strong constraint that makes the next candidate one-hot encoded.

Alternatively, $\mathbf{c} \in \{0, 1\}^m$ is represented as a Boolean vector, where $m$ is the smallest integer which satisfies $2^m \geq N$. Since the optimization of an acquisition function yields real-valued variables, these real values are converted into a Boolean vector by rounding to 0 or 1, before the objective function is evaluated at the corresponding categorical input. In fact, we can interpret this approach as a Bayesian optimization method operating in a 0-1 polytope induced by an injective mapping $\phi : \mathcal{C} \rightarrow \{0, 1\}^m$.

This idea was generalized to solve the online decision making problem in general combinatorial spaces [Rajkumar and Agarwal, 2014], introducing an injective mapping $\phi : \mathcal{C} \rightarrow \mathcal{X}$, where $\mathcal{X} = \mathrm{Conv}(\{\phi(\mathbf{c}) \mid \mathbf{c} \in \mathcal{C}\})$ is a convex polytope and $\mathrm{Conv}(\cdot)$ denotes a convex hull of the given set [Ziegler, 1993]. The low-dimensional online mirror descent algorithm was developed for the case where costs are linear in a low-dimensional vector representation of the decision space [Rajkumar and Agarwal, 2014]. Motivated by [Rajkumar and Agarwal, 2014], we take a similar approach to tackle the combinatorial Bayesian optimization. The main contributions, which are distinct from existing work, are summarized below.

- We use a random mapping for

$$\phi : \mathcal{C} \rightarrow \mathcal{X} = \mathrm{Conv}(\{\phi(\mathbf{c}) \mid \mathbf{c} \in \mathcal{C}\}), \quad (7)$$

  to embed Boolean space into a low-dimensional vector space on which the construction of a probabilistic surrogate model and the optimization of an acquisition function are performed. Such random projection was not considered in [Rajkumar and Agarwal, 2014].

- A random embedding was employed to solve the Bayesian optimization in a very high-dimensional space [Wang et al., 2016], but the combinatorial structure and the corresponding regret analysis were not considered. We show that even with a random projection, our combinatorial Bayesian optimization algorithm achieves a sublinear regret bound with very high probability.

## 4 PROPOSED METHOD

In this section, we present our combinatorial Bayesian optimization algorithm, which is summarized in Algorithm 1. We introduce an injective mapping

$$\phi : \mathcal{C} \rightarrow \widehat{\mathcal{X}}, \quad (8)$$

---

**Algorithm 1** Combinatorial Bayesian Optimization with Random Mapping Functions

**Require:** Combinatorial search space $\mathcal{C}$, time budget $T$, unknown, but observable objective function $f$, mapping function $\phi$.
**Ensure:** Best point $\mathbf{c}_{\mathrm{best}}$.
1: Initialize $\mathbf{C}_1 \subset \mathcal{C}$.
2: Update $\mathbf{y}_1$ by observing $\mathbf{C}_1$.
3: **for** $t = 1, \ldots T$ **do**
4:    Estimate a surrogate function $\widehat{f}(\mathbf{x}; \phi, \mathbf{C}_t, \mathbf{y}_t)$.
5:    Acquire a query point:

$$\mathbf{x}_t^{\dagger} = \arg\max_{\mathbf{x} \in \widehat{\mathcal{X}} \subseteq \mathcal{X}} \widetilde{a}(\mathbf{x}; \widehat{f}(\mathbf{x}; \phi, \mathbf{C}_t, \mathbf{y}_t)). \quad (11)$$

6:    Recover $\mathbf{x}_t^{\dagger}$ to $\mathbf{c}_t^{\dagger}$.
7:    Observe $\mathbf{c}_t^{\dagger}$.
8:    Update $\mathbf{C}_t$ to $\mathbf{C}_{t+1}$ and $\mathbf{y}_t$ to $\mathbf{y}_{t+1}$.
9: **end for**
10: Find the best point and its function value:

$$(\mathbf{c}_{\mathrm{best}}, y_{\mathrm{best}}) = \arg\min_{(\mathbf{c}, y) \in (\mathbf{C}_T, \mathbf{y}_T)} y. \quad (12)$$

11: **return** $\mathbf{c}_{\mathrm{best}}$

---

where $\widehat{\mathcal{X}} = \{\mathbf{x} \mid \mathbf{x} = \phi(\mathbf{c}), \; \forall \mathbf{c} \in \mathcal{C}\}$. With the mapping $\phi$, the black-box optimization problem (1) can be written as

$$\mathbf{c}^{\star} = \arg\min_{\mathbf{x} \in \widehat{\mathcal{X}}} f(\phi^{-1}(\mathbf{x})), \quad (9)$$

where $\phi^{-1}$ is a left inverse of injective function $\phi$.

At iteration $t$, we construct a surrogate model $\widehat{f}(\mathbf{x}; \phi, \mathbf{C}_t, \mathbf{y}_t)$ for $\mathbf{x} \in \mathcal{X}$ using Gaussian process regression on the currently available dataset $\{\mathbf{C}_t, \mathbf{y}_t\} = \{(\mathbf{c}_i, y_i) \mid i = 1, \ldots, t\}$, in order to estimate the underlying objective function $f(\phi^{-1}(\mathbf{x}))$. Then we optimize the acquisition function $\widetilde{a}(\mathbf{x})$ over $\mathcal{X}$ (instead of $\mathcal{C}$), which is built using the posterior mean and variance calculated by the surrogate model $\widehat{f}(\mathbf{x}; \phi, \mathbf{C}_t, \mathbf{y}_t)$, to determine where next to evaluate the objective:

$$\mathbf{x}_t^{\dagger} = \arg\max_{\mathbf{x} \in \widehat{\mathcal{X}} \subseteq \mathcal{X}} \widetilde{a}(\mathbf{x}; \widehat{f}(\mathbf{x}; \phi, \mathbf{C}_t, \mathbf{y}_t)), \quad (10)$$

where $\mathcal{X} = \mathrm{Conv}(\{\phi(\mathbf{c}) \mid \mathbf{c} \in \mathcal{C}\})$. The underlying objective function is evaluated at $\mathbf{c}^{\dagger} = \phi^{-1}(\mathbf{x}^{\dagger})$.

Before we describe the use of random mapping for $\phi$, we summarize a few things that should be noted:

- We assume a mapping $\phi : \mathcal{C} \rightarrow \widehat{\mathcal{X}}$ is injective, so it can be reversed by its left inverse $\phi^{-1} : \widehat{\mathcal{X}} \rightarrow \mathcal{C}$. Note that $|\mathcal{C}| = |\widehat{\mathcal{X}}| = N$.

- We run essential procedures in $\mathcal{X}$, to solve the combinatorial Bayesian optimization, that is a convex polytope

which includes $\widehat{\mathcal{X}}$. Note that $\mathcal{X}$ is supposed to be convex and compact, so is an appropriate feasible decision space for Bayesian optimization [Srinivas et al., 2010].

- Suppose that we are given a positive-definite kernel $k : \mathcal{X} \times \mathcal{X} \to \mathbb{R}$, i.e., $k(\mathbf{x}, \mathbf{x}') = \langle \varphi(\mathbf{x}), \varphi(\mathbf{x}') \rangle_{\mathcal{H}_{\widehat{\mathcal{X}}}}$ where $\varphi(\cdot)$ is a feature map into a RKHS $\mathcal{H}_{\widehat{\mathcal{X}}}$. Then we define a kernel on a RKHS $\mathcal{H}_\mathcal{C}$

$$\widetilde{k}(\mathbf{c}, \mathbf{c}') = \langle \varphi(\phi(\mathbf{c})), \varphi(\phi(\mathbf{c}')) \rangle_{\mathcal{H}_\mathcal{C}}, \quad (13)$$

which is also positive-definite. $\mathcal{H}_\mathcal{C}$ and $\mathcal{H}_{\widehat{\mathcal{X}}}$ are isometrically isomorphic.

- We use $\widetilde{k}$ defined in (13) as a covariance function for Gaussian process regression over $\phi(\mathbf{c})$ which is applied to estimate the surrogate model.

## 4.1 RANDOM MAPPING AND LEFT INVERSE

Now we consider a random mapping $\phi$ on $\mathcal{C}$ into $\mathbb{R}^d$. In practice, we first convert categorical variables $\mathbf{c} \in \mathcal{C}$ into Boolean vectors $\mathbf{b} \in \{0,1\}^m$ [Matoušek, 2002], where $m$ is the smallest integer that satisfies $2^m \geq N$. Then we construct a random matrix $\mathbf{R} \in \mathbb{R}^{d \times m}$ to transform Boolean vectors $\mathbf{b}$ into $d$-dimensional real-valued vectors, i.e.,

$$\phi(\mathbf{c}) = \mathbf{R}\mathbf{b}, \quad (14)$$

where $\mathbf{b}$ is a Boolean vector corresponding to $\mathbf{c}$.

There exists a low-distortion embedding on $\{0,1\}^m$ into $\mathbb{R}^d$ [Trevisan, 2000, Indyk et al., 2017]

We define a random mapping function, which is supported by the theoretical foundations of general random mapping functions. First of all, the existence of transformation from $\mathcal{C}$ to $\mathbb{R}^d$ can be shown. To construct the mapping function that satisfies the aforementioned properties, a combination on $\mathcal{C}$ is transformed to a Boolean vector on $\{0,1\}^m$, where $m$ is the smallest dimensionality that satisfies $2^m \geq N$, which is always possible [Matoušek, 2002]. Then, we can embed $\{0,1\}^m$ into Euclidean space using low-distortion embedding [Trevisan, 2000, Indyk et al., 2017]. Therefore, there exists a transformation from $\mathcal{C}$ to $\mathbb{R}^d$.

Since Bayesian optimization models a surrogate based on similarities between two covariates, we require to define the covariance function defined on $\mathcal{X}$, which is an appropriate representative of the similarities between two inputs on $\mathcal{C}$.

**Lemma 1.** *For $\varepsilon \in (0,1)$ and $d \in \Omega(\log t/\varepsilon^2)$, by the Johnson–Lindenstrauss lemma [Johnson and Lindenstrauss, 1984], Proposition 2.3 in [Trevisan, 2000], and the existence of low-distortion transformation from $\mathcal{C}$ to $\mathbb{R}^d$, a random mapping $\phi : \mathcal{C} \to \mathcal{X}$ preserves a similarity between two any covariates on $\mathcal{C}$:*

$$(1-\varepsilon)\|\mathbf{b} - \mathbf{b}'\|_1 \leq \|\phi(\mathbf{c}) - \phi(\mathbf{c}')\|_2^2 \leq (1+\varepsilon)\|\mathbf{b} - \mathbf{b}'\|_1, \quad (15)$$

*where Boolean vectors $\mathbf{b}, \mathbf{b}' \in \{0,1\}^m$ are equivalent to some $\mathbf{c}, \mathbf{c}' \in \mathcal{C}$ and $t$ is the current iteration.*

*Proof.* Suppose that we have two mapping functions, $\phi_1 : \mathcal{C} \to \{0,1\}^m$ and $\phi_2 : \{0,1\}^m \to \mathbb{R}^d$. A function $\phi_1$ is a bijective function since it maps combinatorial variables to their unique binary variables. Moreover, in Proposition 2.3 in [Trevisan, 2000], $\|\mathbf{b} - \mathbf{b}'\|_p = \|\mathbf{b} - \mathbf{b}'\|_{\text{Hamming}}^{1/p}$ is shown, where $\mathbf{b}, \mathbf{b}' \in \{0,1\}^m \subseteq \mathbb{R}^m$, and $\|\cdot\|_{\text{Hamming}}$ is the Hamming distance. Then, Lemma 1 is readily proved:

$$\begin{aligned} \|\phi(\mathbf{c}) - \phi(\mathbf{c}')\|_2^2 &= \|\phi_2(\mathbf{b}) - \phi_2(\mathbf{b}')\|_2^2 \\ &\geq (1-\varepsilon)\|\mathbf{b} - \mathbf{b}'\|_2^2 \\ &= (1-\varepsilon)\|\mathbf{b} - \mathbf{b}'\|_{\text{Hamming}} \\ &= (1-\varepsilon)\|\mathbf{b} - \mathbf{b}'\|_1, \quad (16) \end{aligned}$$

and

$$\begin{aligned} \|\phi(\mathbf{c}) - \phi(\mathbf{c}')\|_2^2 &= \|\phi_2(\mathbf{b}) - \phi_2(\mathbf{b}')\|_2^2 \\ &\leq (1+\varepsilon)\|\mathbf{b} - \mathbf{b}'\|_2^2 \\ &= (1+\varepsilon)\|\mathbf{b} - \mathbf{b}'\|_{\text{Hamming}} \\ &= (1+\varepsilon)\|\mathbf{b} - \mathbf{b}'\|_1, \quad (17) \end{aligned}$$

by the Johnson–Lindenstrauss lemma, Proposition 2.3 in [Trevisan, 2000], and the existence of low-distortion transformation, where $\phi(\cdot) = \phi_2(\phi_1(\cdot))$. Therefore, by (16) and (17), the following is satisfied:

$$(1-\varepsilon)\|\mathbf{b} - \mathbf{b}'\|_1 \leq \|\phi(\mathbf{c}) - \phi(\mathbf{c}')\|_2^2 \leq (1+\varepsilon)\|\mathbf{b} - \mathbf{b}'\|_1, \quad (18)$$

which concludes the proof of Lemma 1. $\qquad \square$

Lemma 1 encourages us to define a solid mapping function and its adequate inverse, rather than the previous work [Wang et al., 2016]. This method [Wang et al., 2016], which is referred to as REMBO converts from a query point $\mathbf{x}$ to a variable on 0-1 polytope with random matrix in a generative perspective, and then determines a combination with rounding the variable.

Compared to REMBO, we suggest a mapping function from $\mathcal{C}$ to $\mathcal{X}$ with a uniformly random matrix $\mathbf{R} \in \mathbb{R}^{d \times m}$, of which the left inverse is computed by Moore-Penrose inverse $\mathbf{R}^+ \in \mathbb{R}^{m \times d}$, followed by rounding to $\{0,1\}^m$. The Bayesian optimization method using this random matrix and its Moore-Penrose inverse is referred to as *CBO-Recon* from now, because it determines a query combination through reconstruction.

Furthermore, we design a mapping function with a lookup table $\mathbf{L}$, constructed by a uniformly random matrix $\mathbf{R}$. Each row of the table is a key and value pair, where key and value indicate a combination and its embedding vector by $\mathbf{R}$, respectively. After determining a query point on $\mathcal{X}$, we find the closest point in $\mathbf{L}$ and then recover to one of possible combinations on $\mathcal{C}$. We call it as *CBO-Lookup*.

By these definitions and properties, we analyze our algorithm in terms of regrets and discuss more about our method.

# 5 REGRET ANALYSIS

We analyze the regret bound of combinatorial Bayesian optimization, to validate the convergence quality. Following the foundations of the existing theoretical studies [Srinivas et al., 2010, Chowdhury and Gopalan, 2017, Scarlett, 2018], we use Gaussian process upper confidence bound (GP-UCB) as an acquisition function over $\mathbf{x}$:

$$\widetilde{a}(\mathbf{x}; \widehat{f}(\mathbf{x}; \phi, \mathbf{C}_t, \mathbf{y}_t)) = \mu_t(\mathbf{x}) - \beta_t \sigma_t(\mathbf{x}), \quad (19)$$

where $\mathbf{C}_t \in \mathcal{C}^t$, $\mathbf{y}_t \in \mathbb{R}^t$, and $\beta_t$ is a trade-off hyperparameter at iteration $t$. To be clear, by the form of (19), we have to minimize (19) to find the next query point where an objective function is minimized. Before providing the details, the main theorem is described first.

**Theorem 1.** *Let $\delta \in (0,1)$, $\varepsilon \in (0,1)$, and $\beta_t \in \mathcal{O}(\sqrt{\log(\delta^{-1} N t)})$. Suppose that a function $f$ is on a RKHS $\mathcal{H}_\mathcal{C}$ and a kernel $k$ is bounded, $k(\cdot, \cdot) \in [0, k_{max}]$. Then, the kernel-agnostic cumulative regret of combinatorial Bayesian optimization is upper-bounded with a probability at least $1 - \delta$:*

$$R_T \in \mathcal{O}\left(\sqrt{(\sigma_n^2 + \varepsilon^2) N T \log(\delta^{-1} N T)}\right), \quad (20)$$

*where $\sigma_n^2$ is the variance of observation noise and $N$ is the cardinality of $\mathcal{C}$.*

The intuition of Theorem 1 is three-fold. First, it is a sublinear cumulative regret bound, since $\lim_{T \to \infty} R_T / T = 0$, which implies that the cumulative regret does not increase as $T$ goes to infinity. Second, importantly, Theorem 1 shows if $N$ increases, the upper bound of cumulative regret increases. It means that we require additional time to converge the cumulative regret if the cardinality of $\mathcal{C}$ is huge. Moreover, our regret bound is related to a distortion level $\varepsilon^2$ by a random map and an observation noise level $\sigma_n^2$, which implies that small $\varepsilon^2$ and $\sigma_n^2$ yield a lower bound than large values. See the discussion section for the details of this theorem.

## 5.1 ON THEOREM 1

From now, we introduce the lemmas required to proving the main theorems.

**Lemma 2.** *Let $\delta \in (0,1)$ and $\beta_t = \mathcal{O}(\sqrt{\log(\delta^{-1} N t)})$ at iteration $t$. Under Lemma 1 and the properties mentioned above, the following inequality is satisfied:*

$$|f(\mathbf{c}) - \mu_t(\mathbf{x})| \leq \beta_t \sigma_t(\mathbf{x}), \quad (21)$$

*for all $\mathbf{x} \in \widehat{\mathcal{X}}$ where a subset of convex polytope $\widehat{\mathcal{X}}$, with a probability at least $1 - \delta$. In addition, using (21), an instantaneous regret $r_t$ is upper-bounded by $2\beta_t \sigma_t(\mathbf{x}_t^\dagger)$, where $\mathbf{x}_t^\dagger$ is a query point at $t$.*

*Proof.* Lemma 2 is originally proved in [Srinivas et al., 2010], and discussed more in [Scarlett, 2018]; See Lemma 1 in [Scarlett, 2018] for more details. By (21), the upper bound of instantaneous regret can be straightforwardly proved by the definition of GP-UCB and Lemma 2:

$$\begin{aligned} r_t &= f(\phi^{-1}(\mathbf{x}_t^\dagger)) - f(\mathbf{c}^\star) \\ &\leq f(\phi^{-1}(\mathbf{x}_t^\dagger)) - \mu_t(\mathbf{x}_t^\dagger) + \beta_t \sigma_t(\mathbf{x}_t^\dagger) \\ &\leq 2\beta_t \sigma_t(\mathbf{x}_t^\dagger), \end{aligned} \quad (22)$$

which concludes this proof. $\qquad\square$

**Lemma 3.** *Let $\epsilon$ be an observation noise, which is sampled from $\mathcal{N}(0, \sigma_n^2)$. Inspired by [Srinivas et al., 2010], we define a mutual information between noisy observations and function values over $\mathbf{C}_t$,*

$$I(\mathbf{y}_t; \mathbf{f}_t) = I([y_1, \ldots, y_t]; [f(\mathbf{c}_1), \ldots, f(\mathbf{c}_t)]). \quad (23)$$

*Suppose that the mutual information is upper-bounded by the maximum mutual information determined by some $\mathbf{C}^* \subset \mathcal{C}$, which satisfies $|\mathbf{C}^*| = t$:*

$$I(\mathbf{y}_t; \mathbf{f}_t) \leq I(\mathbf{y}^*; \mathbf{f}^*), \quad (24)$$

*where $\mathbf{y}^*$ and $\mathbf{f}^*$ are noisy function values and true function values over $\mathbf{C}^*$, respectively. Then, the mutual information $I(\mathbf{y}_t; \mathbf{f}_t)$ is upper-bounded by the entropy over $\mathbf{y}^\star$,*

$$I(\mathbf{y}_t; \mathbf{f}_t) \leq H(\mathbf{y}^\star), \quad (25)$$

*where $\mathbf{y}^\star$ is noisy function values for $\mathbf{C}^\star \subset \mathcal{C}$ that satisfies $|\mathbf{C}^\star| = N$.*

*Proof.* Given an objective function $f$ and a variance of observation noise $\sigma_n^2$, a mutual information between noisy observations and function values over $\mathbf{C}_t$,

$$I(\mathbf{y}_t; \mathbf{f}_t) = I([y_1, \ldots, y_t]; [f(\mathbf{c}_1), \ldots, f(\mathbf{c}_t)]), \quad (26)$$

where $y_i = f(\mathbf{c}_i) + \epsilon$ and $\epsilon \sim \mathcal{N}(0, \sigma_n^2)$, for $i \in [t]$. As discussed in [Srinivas et al., 2010], the mutual information (26) is upper-bounded by the maximum mutual information determined by some $\mathbf{C}^* \subset \mathcal{C}$, which satisfies $|\mathbf{C}^*| = t$:

$$I(\mathbf{y}_t; \mathbf{f}_t) \leq I(\mathbf{y}^*; \mathbf{f}^*), \quad (27)$$

where $\mathbf{y}^*$ and $\mathbf{f}^*$ are noisy function values and true function values for $\mathbf{C}^*$, respectively. By the definition of mutual information, the following inequality is satisfied:

$$I(\mathbf{y}^*; \mathbf{f}^*) \leq H(\mathbf{y}^*), \quad (28)$$

where $H$ is an entropy. Then, since $H(\mathbf{y}^\star) = H(\mathbf{y}^*) + H(\mathbf{y}^\star \cap (\mathbf{y}^*)^c)$ where $\mathbf{y}^\star$ is noisy function values for $\mathbf{C}^\star = \mathcal{C}$, the mutual information (26) is upper-bounded by the entropy over $\mathbf{y}^\star$,

$$I(\mathbf{y}_t; \mathbf{f}_t) \leq H(\mathbf{y}^\star), \quad (29)$$

which concludes the proof of this lemma. $\qquad\square$

Using the lemmas described above, the proof of Theorem 1 can be provided. For notation simplicity, $\mathbf{x}_t^\dagger$ and $k(\mathbf{x}, \mathbf{x})$ would be written as $\mathbf{x}_t$ and $k_{\max}$, respectively.

*Proof.* A cumulative regret $R_T$ can be described as below:

$$
\begin{aligned}
R_T &= \sum_{t=1}^T r_t \\
&\leq 2\beta_T \sum_{t=1}^T \sigma_t(\mathbf{x}_t) \\
&\leq 2\beta_T \left( T \sum_{t=1}^T \sigma_t^2(\mathbf{x}_t) \right)^{\frac{1}{2}},
\end{aligned}
\tag{30}
$$

by Lemma 2 and the Cauchy-Schwartz inequality. By Lemma 1, we know a covariate $\mathbf{x}_t \in \mathcal{X}$ is distorted within a factor of $(1 \pm \varepsilon)$ from the original space $\mathcal{C}$. Here, from this proposition, we assume $\mathbf{x}_t$ has an input noise, sampled from $\mathcal{N}(\mathbf{0}, \varepsilon^2 \mathbf{I})$. As discussed by McHutchon and Rasmussen [2011], under this assumption and the Taylor's expansion up to the first-order term, the term $\mathbf{K}(\mathbf{X}, \mathbf{X}) + \sigma_n^2 \mathbf{I}$ in the posterior mean and variance functions [Rasmussen and Williams, 2006] is replaced with

$$
\mathbf{K}(\mathbf{X}, \mathbf{X}) + \sigma_n^2 \mathbf{I} + \varepsilon^2 \mathrm{diag}\left( \Delta_{\widehat{f}} \Delta_{\widehat{f}}^\top \right),
\tag{31}
$$

where $\mathbf{X} \in \mathbb{R}^{t \times d}$ is the previously observed inputs, $\Delta_{\widehat{f}} \in \mathbb{R}^{t \times d}$ is a derivative matrix of a surrogate model $\widehat{f}$ over $\mathbf{X}$,[2] and $\mathrm{diag}(\cdot)$ returns a diagonal matrix; See [McHutchon and Rasmussen, 2011] for the details. Using this, (30) is

$$
\begin{aligned}
R_T &\leq 2\beta_T \sqrt{4(\sigma_n^2 + c\varepsilon^2)} \\
&\quad \times \left( T \sum_{t=1}^T \frac{1}{2} \log(1 + (\sigma_n^2 + c\varepsilon^2)^{-1} \sigma_t^2(\mathbf{x}_t)) \right)^{\frac{1}{2}} \\
&\leq 4\beta_T \sqrt{(\sigma_n^2 + c\varepsilon^2) T \, I(\mathbf{y}^*; \mathbf{f}^*)} \\
&\leq 4\beta_T \sqrt{(\sigma_n^2 + c\varepsilon^2) T \, H(\mathbf{y}^\star)},
\end{aligned}
\tag{32}
$$

where $c = \max(\mathrm{diag}(\Delta_{\widehat{f}} \Delta_{\widehat{f}}^\top))$, by Lemma 3, Lemma 5.3 in [Srinivas et al., 2010], and $\alpha \leq 2\log(1 + \alpha)$ for $0 \leq \alpha \leq 1$. Finally, by the definition of entropy and the Hadamard's inequality, (32) is expressed as

$$
\begin{aligned}
R_T &\leq 4\beta_T \sqrt{(\sigma_n^2 + c\varepsilon^2) T \log |2\pi e \Sigma|} \\
&\leq 4\beta_T \sqrt{(\sigma_n^2 + c\varepsilon^2) NT \log(2\pi e k_{\max})} \\
&\in \mathcal{O}\left( \beta_T \sqrt{(\sigma_n^2 + \varepsilon^2) NT} \right) \\
&= \mathcal{O}\left( \sqrt{(\sigma_n^2 + \varepsilon^2) NT \log(\delta^{-1} NT)} \right),
\end{aligned}
\tag{33}
$$

where $N$ is the cardinality of $\mathcal{C}$ and $k_{\max}$ is the maximum value of kernel $k$. $\qquad\square$

---

[2] Gaussian process is at least once differentiable with at least once differentiable kernels.

# 6   DISCUSSION

In this section we provide more detailed discussion on combinatorial Bayesian optimization and our methods.

**Acquisition function optimization.** Unlike the existing methods and other baselines, CBO-Recon and CBO-Lookup should find a query point on convex polytope $\mathcal{X}$. If a random matrix $\mathbf{R}$ is constructed by a distribution supported on a bounded interval, $\mathcal{X}$ is naturally compact. Thus, CBO-Recon and CBO-Lookup optimize an acquisition function on the convex space determined by $\mathbf{R}$.

**Rounding to binary variables.** To recover a covariate on $\mathbb{R}^d$ to $\mathcal{C}$, rounding to binary variables is required in REMBO and CBO-Recon as well as other existing methods. These methods must choose a proper threshold between 0 and 1 after scaling them between that range [Hutter et al., 2011, Wang et al., 2016], but choosing a rule-of-thumb threshold is infeasible because an objective is unknown and we cannot validate the threshold without direct access to the objective. On the contrary, CBO-Lookup does not require any additional hyperparameter for rounding to binary variables.

$L_1$ **regularization for sparsity.** As proposed and discussed in [Baptista and Poloczek, 2018, Oh et al., 2019], the objective for combinatorial optimization is penalized by $L_1$ regularization. This regularization technique is widely used in machine learning in order to induce a sparsity [Tibshirani, 1996] and avoid over-fitting. In this discussion, we rethink this sparsification technique for combinatorial structure. A representation on the Hamming space is not always sparse, because, for example, a solution can be $[1, 1, \ldots, 1]$, which is not sparse.

**Complexity analysis.** The time complexity of common Bayesian optimization is $\mathcal{O}(T^3)$ where $T$ is the number of iterations (i.e., we assume the number of observations is proportional to $T$, because it evaluates a single input at every iteration), due to the inverse of covariance matrix. Our CBO-Lookup holds the time complexity of common Bayesian optimization, and moreover with the Cholesky decomposition it can be reduced to $\mathcal{O}(T^3/6)$ [Rasmussen and Williams, 2006]. Compared to BOCS proposed by Baptista and Poloczek [2018], our algorithm has lower complexity than the BOCS variants. The CBO-Lookup algorithm requires the space complexity $\mathcal{O}(md)$, because it constructs a lookup table. However, we can speed up this pre-processing stage by preemptively constructing the table before the optimization process and searching the table with a hash function [Nayebi et al., 2019]. In practice, the combinatorial space of the order of $2^{24}$ requires less than 2 GB.

**Regret bound.** As described in the analysis section, Theorem 1 supports our intuition in terms of $\varepsilon^2$, $\sigma_n^2$, $\delta$, $N$, and $T$. However, since we prove the cumulative regret bound

for any kernel that is bounded and at least once differentiable, it is considered as a loose bound. According to other theoretical studies [Srinivas et al., 2010, Chowdhury and Gopalan, 2017, Scarlett, 2018], this kernel-agnostic bound is readily expanded into a tighter bound, assuming a specific kernel. To focus on the problem of combinatorial Bayesian optimization itself and moderate the scope of this work, the analysis for tighter bound will be left to the future work.

# 7 EXPERIMENTS

We test the experiments on combinatorial optimization: thumbs-up maximization, seesaw equilibrium, binary quadratic programming, Ising model sparsification, and subset selection. We first introduce the experimental setups.

## 7.1 EXPERIMENTAL SETUP

To optimize objectives over combinations, we use the following combinatorial Bayesian optimization strategies:

- Random: It is a simple, but practical random search strategy [Bergstra and Bengio, 2012].

- Bin-AA: It uses a surrogate model with AA kernel [Aitchison and Aitken, 1976], over binary variables.

- Bin-Round: The surrogate model is defined on $[0, 1]^m$ and acquired points are determined by rounding. The threshold for rounding is $0.5$.

- Dec-Round: Binary representation is converted to a decimal number. Then, common Bayesian optimization is conducted on $[0, 2^m - 1]$.

- SMAC: It is a sequential model-based optimization with random forests [Hutter et al., 2011]. It optimizes on binary space.

- BOCS-SA/BOCS-SDP: They are proposed by Baptista and Poloczek [2018]. We use their official implementation and follow their settings. Unless otherwise specified, $\lambda$ is set to zero.

- COMBO: This approach [Oh et al., 2019] constructs a surrogate function on the space of Cartesian product of graphs. Each graph corresponds to a categorical variable and a diffusion kernel over graphs is used to measure the similarity between combinations.

- REMBO/CBO-Recon: These strategies find the next query point on $\mathbb{R}^d$ and recover it to a combination with rounding. To round to a binary variable, thresholds for REMBO and CBO-Recon are set to $0.25$ and $0.02$, respectively. If it is not specified, we use $d = 20$.

- CBO-Lookup: It determines the next query point after mapping a combination to $\mathbb{R}^d$, and recover it to a combination with a pre-defined lookup table. We use the same setting of REMBO and CBO-Recon for $d$.

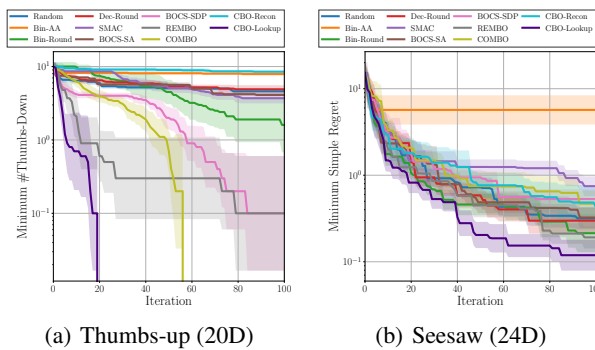

(a) Thumbs-up (20D)   (b) Seesaw (24D)

Figure 1: Bayesian optimization results on thumbs-up maximization and seesaw equilibrium. All the experiments are repeated 10 times.

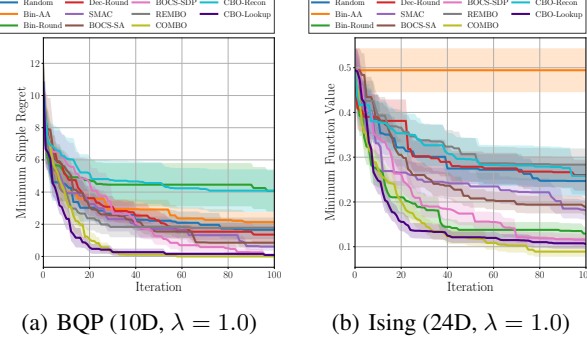

(a) BQP (10D, $\lambda = 1.0$)  (b) Ising (24D, $\lambda = 1.0$)

Figure 2: Bayesian optimization results on binary quadratic programming and Ising model sparsification. All the experiments are repeated 10 times.

To compare all the methods fairly, we set the same initializations across all the methods, by fixing a random seed for the same runs. The number of initializations is set to 1 for most of experiments. Gaussian process regression with Matérn 5/2 kernel is employed as a surrogate model for Bin-Round, Dec-Round, REMBO, CBO-Recon, and CBO-Lookup. The GP-UCB acquisition function is used as an acquisition function for Bin-AA, REMBO, CBO-Recon, and CBO-Lookup. For COMBO, we use the official implementation provided by Oh et al. [2019], but we modify some setups, e.g., initial points, for fair comparisons. The expected improvement criterion is applied for Bin-Round and Dec-Round. The hyperparameters of Gaussian process regression are optimized by marginal likelihood maximization. All the experiments are repeated 10 times unless otherwise specified.

## 7.2 THUMBS-UP MAXIMIZATION

This experiment is an artificial circumstance to maximize the number of thumbs-up where the total number of thumbs-up and thumbs-down is fixed as $m$. As shown in Figure 1(a),

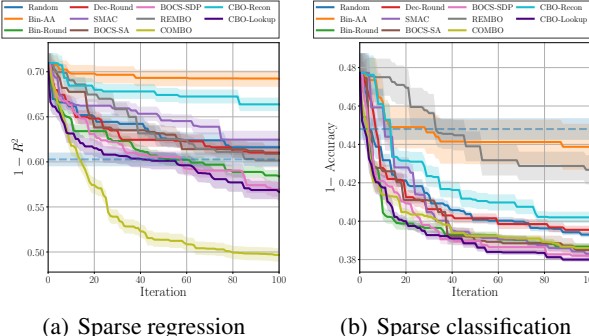

(a) Sparse regression      (b) Sparse classification

Figure 3: Results on subset selection, where $\rho = 0.01$ and $\nu = 1.0$. Dotted lines are the ones with $L_1$ regularization. All the experiments are repeated 10 times.

CBO-Lookup performs well compared to other methods.

## 7.3 SEESAW EQUILIBRIUM

It is a combinatorial problem for balancing the total torques of weights on a seesaw, which has $m/2$ weights on the left side of seesaw and the other $m/2$ weights on the right side, where $m$ is an even number. The weight $w_i$ is uniformly chosen from $[0.5, 2.5]$ for all $i \in \{0, \ldots, m-1\}$. Each torque for $w_i$ is computed by $r_i w_i$ where $r_i$ is the distance from the pivot of seesaw, and we set a distance between $w_i$ and the pivot as $r_i = i - m/2$ for the weights on the left side and $r_i = i - m/2 + 1$ for the weights on the right side. As presented in Figure 1(b), CBO-Lookup achieves the best result in this experiment.

## 7.4 BINARY QUADRATIC PROGRAMMING

This experiment is to minimize an objective of binary quadratic programming problem with $L_1$ regularization,

$$f(\mathbf{c}) = \phi(\mathbf{c})^\top \mathbf{Q} \phi(\mathbf{c}) + \lambda \|\phi(\mathbf{c})\|_1, \qquad (34)$$

where $\phi$ is a mapping function to binary variables, $\mathbf{Q} \in \mathbb{R}^{m \times m}$ is a random matrix, and $\lambda$ is a coefficient for $L_1$ regularization. We follow the settings proposed by [Baptista and Poloczek, 2018]. Similar to thumbs-up maximization and seesaw equilibrium, CBO-Lookup shows consistent performance compared to other baselines; see Figure 2(a).

## 7.5 ISING MODEL SPARSIFICATION

This problem finds an approximate distribution $q(\phi(\mathbf{c}))$, which is close to

$$p(\phi(\mathbf{c})) = \frac{\exp(\phi(\mathbf{c})^\top J^p \phi(\mathbf{c}))}{\sum_{\mathbf{c}'} \exp(\phi(\mathbf{c}')^\top J^p \phi(\mathbf{c}'))}, \qquad (35)$$

where $\phi$ is a mapping function to $\{-1, 1\}^m$ and $J^p$ is a symmetric interaction matrix for Ising models. To find $q(\phi(\mathbf{c}))$, we compute and minimize the Kullback-Leibler divergence between two distributions in the presence of $L_1$ regularization. Our implementation for this experiment refers to the open-source repository of [Baptista and Poloczek, 2018]. CBO-Lookup is comparable to or better than other methods, as shown in Figure 2(b).

## 7.6 SUBSET SELECTION

A regularization technique such as Lasso [Tibshirani, 1996] has been widely used to solve this problem, where some variables are less correlated to the other variables, but in this experiment we solve this variable selection using combinatorial Bayesian optimization. Our target tasks are categorized as two classes: (i) a sparse regression task, i.e., Figure 3(a), and (ii) a sparse classification task, i.e., Figure 3(b) [Hastie et al., 2015]. For a fair comparison, we split the datasets into training, validation, and test datasets. The validation dataset is used to select variables using Bayesian optimization, and the evaluation for the selected variables of test dataset is reported.

To conduct the Bayesian optimization methods on a sparse regression task, we follow the data creation protocol described in [Hastie et al., 2017]. Given the number of data $n$, the dimension of data $p$, a sparsity level $s$, a predictor correlation level $\rho$, a signal-to-noise ratio level $\nu$, our regression model is defined as $\mathbf{y} = \mathbf{X}\boldsymbol{\beta}$, where $\mathbf{y} \in \mathbb{R}^n$, $\mathbf{X} \in \mathbb{R}^{n \times p}$, and $\boldsymbol{\beta} \in \mathbb{R}^p$. We use beta-type 2 [Bertsimas et al., 2016] with $s$ for $\boldsymbol{\beta}$ and draw $\mathbf{X}$ from $\mathcal{N}(\mathbf{0}, \boldsymbol{\Sigma})$ where $[\boldsymbol{\Sigma}]_{ij} = \rho^{|i-j|}$ for all $i, j \in [p]$. Then, we draw $\mathbf{y}$ from $\mathcal{N}(\mathbf{X}\boldsymbol{\beta}, \sigma^2 \mathbf{I})$, where $\sigma^2 = (\boldsymbol{\beta}^\top \boldsymbol{\Sigma} \boldsymbol{\beta})/\nu$.

For a sparse classification, we employ Mobile Price Classification dataset [Sharma, 2018], of which the variables are less correlated. The base classifier for all the experiments are support vector machines and the variables selected by Bayesian optimization are given to train and test the dataset. As presented in Figure 3(b), our method shows the better result than other methods in this experiment.

**Comparisons with COMBO.** As is generally known in the Bayesian optimization community, COMBO [Oh et al., 2019] shows consistent performance in the experiments run in this work. Since COMBO uses graph-structured representation as the representation of categorical variables, it directly seeks an optimal combination on graphs by considering the connectivity between variables. This capability allows us to effectively find the optimum. Compared to the known sophisticated COMBO method, CBO-Lookup also shows consistent performance across various tasks as shown in Figure 1, Figure 2(a), Figure 2(b), and Figure 3(b), and guarantees a sublinear cumulative regret bound with a low-distortion embedding as shown in Theorem 1.

Moreover, our method is beneficial for cheap computational complexity and easy implementation, by utilizing any off-the-shelf implementation of vector-based Bayesian optimization. In addition to these, without loss of generality, our method is capable of optimizing a black-box objective on a high-dimensional mixed-variable space by embedding a high-dimensional mixed-variable space to a low-dimensional continuous space. Unfortunately, it is unlikely that COMBO can solve the optimization problem defined on a high-dimensional mixed-variable space.

# 8 CONCLUSION

In this work, we analyze the aspect of combinatorial Bayesian optimization and highlight the difficulty of this problem formulation. Then, we propose a combinatorial Bayesian optimization method with a random mapping function, which guarantees a sublinear cumulative regret bound. Our numerical results demonstrate that our method shows satisfactory performance compared to other existing methods in various combinatorial problems.

**Author Contributions**

J. Kim led this work, suggested the main idea of this paper, implemented the proposed methods, and wrote the paper. S. Choi and M. Cho are co-corresponding authors and the order of them is randomly determined. S. Choi and M. Cho advised this work, improved the proposed methods, and wrote the paper. Along with these contributions, S. Choi introduced the fundamental backgrounds of this work, and M. Cho substantially revised the paper.

**Acknowledgements**

This work was supported by Samsung Research Funding & Incubation Center of Samsung Electronics under Project Number SRFC-TF2103-02.

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
