# OpenReview forum: "Combinatorial Bayesian Optimization with Random Mapping Functions to Convex Polytopes"
_auai.org/UAI/2022/Conference — UAI 2022 Poster_

### Official Review · Reviewer_bun6 · 2022-03-23

**Q2(1) Originality/Novelty:** 3
**Q2(2) Significance/Impact:** 3
**Q2(3) Correctness/Technical Quality:** 3
**Q2(6) Clarity Of Writing:** 4
**Q6 Overall Score:** 7
**Q8 Confidence In Your Score:** 3

**Q1 Summary And Contributions:**

The paper presents an approach for combinatorial optimization based on a random map to a continuous space. The approach is shown to have sublinear cumulative regret with high probability and demonstrated to have good/competitive performance compared to other optimization methods on a variety of optimization problems.

**Q10 Ethical Concerns (Optional):**

No.

**Q2 Assessment Of The Paper:**

More detailed information regarding each of these aspects is given below:

**Q2(4) Quality Of Experiments (Optional):**

4: Excellent: The experimental evaluation is comprehensive and the results are compelling.

**Q2(5) Reproducibility:**

2: Fair: Key resources (e.g., proofs, code, data) are unavailable but key details (e.g., proof sketches, experimental setup) are sufficiently well-described for an expert to confidently reproduce the main results.

**Q3 Main Strengths:**

* The paper is very well-written.
* The presented method is justified by theoretical results.
* The experimental results show an improvement over the state of the art across many optimization problems.

**Q4 Main Weakness:**

No code was submitted, although the algorithm is described quite clearly in the paper.

**Q5 Detailed Comments To The Authors:**

1. Why is combinatorial explosion a problem for the binary encoding of categorical data? How come the n -> log n improvement is insufficient?
2. The proof of Lemma 1 states that \phi_1 is bijective, but it can only be bijective if the cardinality of \mathcal{C} is 2^m.
3. On page 6, you write that \mathcal{O}(T^3) can be reduced to \mathcal{O}(T^3/6), but \mathcal{O}(T^3) = \mathcal{O}(T^3/6).
4. In the figures, Random and CBO-Lookup have the same colour. The figures could also benefit from line types in addition to colours.

**Q7 Justification For Your Score:**

I think the paper contributes a well-justified approach that is likely to be useful across a range of optimization problems. The paper has no major weaknesses, reads well, and contains both theoretical and experimental results that support the approach and its applicability.

**Q9 Complying With Reviewing Instructions:**

1: Yes.

---

### Official Review · Reviewer_XMmV · 2022-04-09

**Q2(1) Originality/Novelty:** 3
**Q2(2) Significance/Impact:** 3
**Q2(3) Correctness/Technical Quality:** 3
**Q2(6) Clarity Of Writing:** 3
**Q6 Overall Score:** 6
**Q8 Confidence In Your Score:** 4

**Q1 Summary And Contributions:**

The authors propose a method for Bayesian optimization in combinatorial space to make the optimization more efficient. The authors propose at each step of Bayesian optimization to embed the categorical inputs to lower-dimensional representation by random projection, perform the optimization in low dimensional space, and then revert back to the original space for function evaluation before starting the next step of Bayesian optimization.

**Q2 Assessment Of The Paper:**

More detailed information regarding each of these aspects is given below:

**Q2(4) Quality Of Experiments (Optional):**

3: Good: The experimental evaluation is adequate, and the results convincingly support the main claims.

**Q2(5) Reproducibility:**

2: Fair: Key resources (e.g., proofs, code, data) are unavailable but key details (e.g., proof sketches, experimental setup) are sufficiently well-described for an expert to confidently reproduce the main results.

**Q3 Main Strengths:**

The paper is well written, the authors clearly describe and identify the gap in current literature. Comprehensive experiments are carried out to show empirical evidence that the proposed technique has superior performance. Theoretical analysis is provided to show the optimization guarantee of the proposed algorithm.

**Q4 Main Weakness:**

The figures are sometimes too crowded, the methodology can be described more clearly, and experimental details can be expanded. Detailed comments provided later.

**Q5 Detailed Comments To The Authors:**

The authors propose a method for Bayesian optimization in combinatorial space to make the optimization more efficient. Bayesian optimization with the Gaussian process works with continuous inputs but struggles to handle categorical inputs. Typically, these categorial inputs are converted to high-dimensional one-hot encoding, which makes the optimization inefficient. The authors propose at each step of Bayesian optimization to embed the categorical inputs to lower-dimensional representation by random projection, perform the optimization in low dimensional space, and then revert back to the original space for function evaluation before starting the next step of Bayesian optimization. The performance of the proposed technique is shown empirically, and theoretical analysis is provided about the optimization guarantee.

The contribution that the authors present can be clarified, for example in the introduction CBO-Lookup is mentioned while later CBO-Recon is also mentioned. The construction of the lookup table which is part of the CBO-Lookup is also not clear, are the authors using random projections of the already observed data points, in which case how is exploration happening? If there are additional points being stored in the lookup table the details are not clear on that. Figures 1,2,3 showing experimental evaluations are very crowded and sometimes require significant attention from the reader. The details of the experiments as in sections 7.2-7.5 could be elaborated as it is hard to understand from the figures what are the metrics, and how are they significant. It seems also the proposed method performs suboptimally in figure 3a, but no details or comments are made to address that.

**Q7 Justification For Your Score:**

The authors present their idea clearly in light of the gaps in current literature, however, there are details that can be clarified furthermore.

**Q9 Complying With Reviewing Instructions:**

1: Yes.

---

### Official Review · Reviewer_x7Si · 2022-04-12

**Q2(1) Originality/Novelty:** 2
**Q2(2) Significance/Impact:** 3
**Q2(3) Correctness/Technical Quality:** 3
**Q2(6) Clarity Of Writing:** 3
**Q6 Overall Score:** 6
**Q8 Confidence In Your Score:** 4

**Q1 Summary And Contributions:**

Bayesian optimization with categorical inputs is a challenging problem. Previous approaches tried to solve it with binary encoding or one-hot encoding. However, they both suffer either constrained optimization issues or feasibility issues. The authors proposed to use random mapping to embed boolean space into low-dimensional vector space. They provide theoretical justifications for the proposed method and did a regret analysis.

**Q2 Assessment Of The Paper:**

More detailed information regarding each of these aspects is given below:

**Q2(4) Quality Of Experiments (Optional):**

3: Good: The experimental evaluation is adequate, and the results convincingly support the main claims.

**Q2(5) Reproducibility:**

3: Good: Key resources (e.g., proofs, code, data) are available and key details (e.g., proofs, experimental setup) are sufficiently well-described for competent researchers to confidently reproduce the main results.

**Q3 Main Strengths:**

The authors give a theoretical analysis of the proposed method which is good. The authors also have done quite comprehensive experiments to prove the effectiveness of the proposed method.

**Q4 Main Weakness:**

Random sampling has been applied to BO tasks to resolve high-dimensional inputs, though less combinatorial search analysis and the novelty of this paper are not significant.

**Q5 Detailed Comments To The Authors:**

What are the empirical time and space analyses with real-world applications?

**Q7 Justification For Your Score:**

The study by the authors is comprehensive and the experiments are done are fairly well. Though the novelty is not that significant. Overall this is an interesting work and can benefit the BO community.

**Q9 Complying With Reviewing Instructions:**

1: Yes.

---

### Official Review · Reviewer_wfn1 · 2022-04-16

**Q2(1) Originality/Novelty:** 3
**Q2(2) Significance/Impact:** 3
**Q2(3) Correctness/Technical Quality:** 3
**Q2(6) Clarity Of Writing:** 3
**Q6 Overall Score:** 6
**Q8 Confidence In Your Score:** 3

**Q1 Summary And Contributions:**

The authors present a method for Bayesian optimization in a combinatorial space. The main idea is to use a random mapping which embeds the combinatorial space into a convex polytope in a continuous space, on which the BO methods are run to determine a solution to the black-box optimization in the combinatorial space. The authors present a regret analysis for the proposed method and experiments demonstrate satisfactory performance compared to existing methods.


**Q2 Assessment Of The Paper:**

More detailed information regarding each of these aspects is given below:

**Q2(4) Quality Of Experiments (Optional):**

3: Good: The experimental evaluation is adequate, and the results convincingly support the main claims.

**Q2(5) Reproducibility:**

3: Good: Key resources (e.g., proofs, code, data) are available and key details (e.g., proofs, experimental setup) are sufficiently well-described for competent researchers to confidently reproduce the main results.

**Q3 Main Strengths:**

- Relevant problem.
- Well written paper.
- Exhaustive experimental results.
- Comparison to a significant number of baselines.
- Theoretical results for regret analysis.



**Q4 Main Weakness:**

- One of the proposed methods CBO-Recon performs rather poorly, but little analysis is done regarding this.



**Q5 Detailed Comments To The Authors:**

- It would be good if the authors could comment more on the poor performance of CBO-Recon.



**Q7 Justification For Your Score:**

This a clearly wirtten paper, describing a simple method that performs well in practice as shown in exhaustive experiments.

**Q9 Complying With Reviewing Instructions:**

1: Yes.

---

### Decision · Program_Chairs · 2022-05-15

**Decision:**

Accept (Poster)

**Comment:**

Meta Review: The authors present a technique for applying Bayesian optimization to combinatorial spaces, in which the discrete space is embedded in a continuous space through a random mapping, on which a Gaussian process method for Bayesian optimization can be applied.  They then map the acquisition function's optimum back to the combinatorial space for evaluation.  The paper provides a regret analysis of the overall approach, and evaluations in several experimental domains.

Overall, reviewers were positive about the work.  Specific reviews contain several suggestions for improving clarity of the presentation.  While many parts of the work are similar to prior approaches, the problem itself appears to be different enough to be novel and interesting.